# Neural Network Approximation of Lipschitz Functions in High Dimensions with Applications to Inverse Problems

## Abstract

The remarkable successes of neural networks in a huge variety of inverse problems have fueled their adoption in disciplines ranging from medical imaging to seismic analysis over the past decade. However, the high dimensionality of such inverse problems has simultaneously left current theory, which predicts that networks should scale exponentially in the dimension of the problem, unable to explain why the seemingly small networks used in these settings work as well as they do in practice. To reduce this gap between theory and practice, a general method for bounding the complexity required for a neural network to approximate a Lipschitz function on a high-dimensional set with a low-complexity structure is provided herein. The approach is based on the observation that the existence of a linear Johnson-Lindenstrauss embedding $A \in \mathbb{R}^{d \times D}$ of a given high-dimensional set $\mathcal{S} \subset \mathbb{R}^D$ into a low dimensional cube $[-M, M]^d$ implies that for any Lipschitz function $f : \mathcal{S} \to \mathbb{R}^p$, there exists a Lipschitz function $g : [-M, M]^d \to \mathbb{R}^p$ such that $g(Ax) = f(x)$ for all $x \in \mathcal{S}$. Hence, if one has a neural network which approximates $g : [-M, M]^d \to \mathbb{R}^p$, then a layer can be added which implements the JL embedding $A$ to obtain a neural network which approximates $f : \mathcal{S} \to \mathbb{R}^p$. By pairing JL embedding results along with results on approximation of Lipschitz functions by neural networks, one then obtains results which bound the complexity required for a neural network to approximate Lipschitz functions on high dimensional sets. The end result is a general theoretical framework which can then be used to better explain the observed empirical successes of smaller networks in a wider variety of inverse problems than current theory allows.

## 1 Introduction

At present various network architectures (NN, CNN, ResNet) achieve state-of-the-art performance in a broad range of inverse problems, including matrix completion (Zheng *et al.*, 2016; Monti *et al.*, 2017; Dziugaite & Roy, 2015; He *et al.*, 2017) image-deconvolution (Xu *et al.*, 2014; Kupyn *et al.*, 2018), low-dose CT-reconstitution (Nah *et al.*, 2017), electric and magnetic inverse Problems (Coccorese *et al.*, 1994) (seismic analysis, electromagnetic scattering). However, since these problems are very high dimensional, classical universal approximation theory for such networks provides very pessimistic estimates of the network sizes required to learn such inverse maps (i.e., as being much larger than what standard computers can store, much less train). As a result, a gap still exists between the widely observed successes of networks in practice and the network size bounds provided by current theory in many inverse problem applications. The purpose of this paper is to provide a refined bound on the size of networks in a wide range of such applications and to show that the network size is indeed affordable in many inverse problem settings. In particular, the bound developed herein depends on the model complexity of the domain of the forward map instead of the domain's extrinsic input dimension, and therefore is much smaller in a wide variety of model settings.

To be more specific, recall in most inverse problems one aims to recover some signal $x$ from its measurement $y = F(x)$. Here $y$ and $x$ could both be high dimensional vectors, or even matrices and tensors, and $F$, which is called the forward map/operator, could either be linear or nonlinear with various regularity conditions depending on the application. In all cases, however, recovering $x$ from $y$ amounts to inverting $F$. In other words, one aims want to find the operator $F^{-1}$, that sends every

measurement $y$ back to the original signal $x$. Depending on the specific application of interest, there are various commonly considered forms of the forward map $F$. For example, $F$ could be a linear map from high to low dimensions as in compressive sensing applications; $F$ could be a convolution operator that computes the shifted local blurring of an image as in the image deblurring setting; $F$ could be a mask that filters out the unobserved entries of the data as in the matrix completion application; or $F$ could also be the source-to-solution map of a differential equation as in ODE/PDE based inverse problems.

In most of these applications, the inverse operator $F^{-1}$ does not possess a closed-form expression. As a result, in order to approximate the inverse one commonly uses analytical approaches that involve solving, e.g., an optimization problem. Take the sparse recovery as an example. With the prior knowledge that the true signal $x \in \mathbb{R}^n$ is sparse, one can recover it from the under-determined measurements $\mathbb{R}^m \ni y = Ax$ with $m < n$) by solving the optimization problem

$$\hat{x} = \arg\min_z \|z\|_0, \quad Az = y$$

The inverse of the linear measurement map $F(x) = y = Ax$ when restricted to the low-complexity domain of sparse vectors has an inverse, $F^{-1}(y)$, that is then the minimizer $\hat{x}$ above.

Note that traditional optimization-based approaches could be extremely slow for large-scale problems (e.g., for $n$ large above). Alternatively, we can approximate the inverse operator by a neural network instead. Amortizing the initial cost of an expensive training stage, the network can later achieve unprecedented speed over time at the test stage leading to better total efficiency over its lifetime. To realize this goal, however, we need to first find a neural network architecture $f_\theta$, and train it to approximate $F^{-1}$, so that the approximation error $\max_y \|f_\theta(y) - F^{-1}(y)\| = \|f_\theta(y) - x\|$ is small. The purpose of this paper is to provide a unified way to give a meaningful estimation of the size of the network that one can use to set up the network in situations where the domain of $F$ is low-complexity as is the case in, e.g., compressive sensing, low-rank matrix completion, deblurring with low-dimensional signal assumptions, etc..

## 2 RELATED WORK

The expressive power of neural networks is important in applications as a means of both guiding network architecture design choices, as well as for providing confidence that good network solutions exist in general situations. As a result, numerous results about the approximation power has been established in recent years (Zhou, 2020; Petersen & Voigtlaender, 2020; Yarotsky, 2022; 2018; Lin & Jegelka, 2018). Most results concern the approximation of functions on $\mathbb{R}^D$, however, and yield network sizes that increase exponentially with the input dimension $D$. As a result, the high dimensionality of many inverse problems leads to bounds from most of the existing literature which are too large to explain the observed empirical success of neural approaches in such applications.

A similar high-dimensional scaling issue arises in many image classification tasks as well. Motivated by this setting (Chen *et al.*, 2019) refined previous approximation results for ReLU networks, and showed that input data that is close to a low-dimensional manifold leads to network sizes that only grow exponentially with respect to the intrinsic dimension of the manifold. However, this improved bound relies on the data fitting manifold assumption which is quite strong in the inverse problems setting. For example, even the "simple" sparse recovery problem does not have a domain/range that forms a manifold (note that the intersections of $s$-dimensional subspaces prevent from it being a manifold). Therefore, to study expressive power of networks on inverse problems needs to remove such strict manifold assumptions. Another mild issue with such manifolds results is that the number of neurons also depends on the curvature of the manifold in question which can be difficult to estimate. Furthermore, such curvature dependence is unavoidable for manifold results and needs to be incorporated into any valid bounds.[1]

In this paper, we provide another way to estimate the size of the network, by directly using the Guassian width of the data as a measure of its inherent complexity. Our result can therefore be

---

[1] To see why, e.g., curvature dependence is unavoidable, consider any discrete training dataset in a compact ball. There always exists a 1-dimensional manifold, namely a curve, that goes through all the data points. Thus, the mere existence of the 1-dimensional manifold does not mean the data complexity is low. Curvature information and other manifold properties matter as well!

considered generalization of the manifold result discussed above in two ways. First, it applies to more arbitrary data sets with low complexities. And, it also applies to other types of networks besides just feedforward ReLu networks. Both types of generalization are then shown to be useful and applicable to various inverse problems.

## 3  MAIN RESULTS

We begin by stating a few definitions. We say that a neural network $\epsilon$-approximates a function $f$ if the function implemented by the neural network $\widehat{f}$ satisfies $\|\widehat{f}(\boldsymbol{x}) - f(\boldsymbol{x})\|_\infty \leq \epsilon$ for all $\boldsymbol{x}$ in the domain of $f$. We say that a neural network architecture $\epsilon$-approximates any function in a function class $\mathcal{F}$ if for any function $f \in \mathcal{F}$, there exists a choice of edge weights such that the function $\widehat{f}$ implemented by the neural network with that choice of edge weights satisfies $\|\widehat{f}(\boldsymbol{x}) - f(\boldsymbol{x})\|_\infty \leq \epsilon$ for all $\boldsymbol{x}$ in the domain of $f$.

Also, for any positive integers $d < D$, any set $\mathcal{S} \subset \mathbb{R}^D$, and any constant $\rho \in (0,1)$, we say that a matrix $\boldsymbol{A} \in \mathbb{R}^{d \times D}$ is a $\rho$-JL (Johnson-Lindenstrauss) embedding of $\mathcal{S}$ if

$$(1-\rho)\|\boldsymbol{x} - \boldsymbol{x}'\|_2 \leq \|\boldsymbol{A}\boldsymbol{x} - \boldsymbol{A}\boldsymbol{x}'\|_2 \leq (1+\rho)\|\boldsymbol{x} - \boldsymbol{x}'\|_2 \quad \text{for all} \quad \boldsymbol{x}, \boldsymbol{x}' \in \mathcal{S}.$$

If we furthermore have $\boldsymbol{A}(\mathcal{S}) := \{\boldsymbol{A}\boldsymbol{x} : \boldsymbol{x} \in \mathcal{S}\} \subset \mathcal{T}$, we say that $\boldsymbol{A}$ is a $\rho$-JL embedding of $\mathcal{S}$ into $\mathcal{T}$. Intuitively, a $\rho$-JL embedding of $\mathcal{S}$ into $\mathbb{R}^d$ transforms $\mathcal{S}$ from a high-dimensional space to a low-dimensional space without significantly distorting distances between points.

**Contributions**: Existing universal approximation theorems for various types of neural networks are mainly stated for functions defined on an $d$-dimensional cube. Our main contribution is to generalize these results to functions defined on arbitrary JL-embedable sets, which possibly reside in very high dimensions. We then demonstrate how our result can be applied to inverse problems to obtain a reasonable estimate of the network size.

Since our theory is to be applied to general inverse problems, for which we cannot assume anything more than Lipschitz continuous. Hence in this paper, we focus on the class of Lipschitz functions.

More explicitly, we show that if there exists a $\rho$-JL embedding of a high-dimensional set $\mathcal{S} \subset \mathbb{R}^D$ into a low-dimensional cube $[-M, M]^d$, then we can use any neural network architecture which can $\epsilon$-approximate $\frac{L}{1-\rho}$-Lipschitz functions on $[-M, M]^d$ to construct a neural network architecture which can $\epsilon$-approximate $L$-Lipschitz functions on $\mathcal{S}$. To establish this, we show that if there exists $\rho$-JL embedding $\boldsymbol{A} \in \mathbb{R}^{d \times D}$ of $\mathcal{S} \subset \mathbb{R}^D$ into $d$-dimensions, then for any $L$-Lipschitz function $f : \mathcal{S} \to \mathbb{R}^p$, there exists a $\frac{L}{1-\rho}$-Lipschitz function $g : [-M, M]^d \to \mathbb{R}^p$ (where $M = \sup_{\boldsymbol{x} \in \mathcal{S}} \|\boldsymbol{A}\boldsymbol{x}\|_\infty$) such that $g(\boldsymbol{A}\boldsymbol{x}) = f(\boldsymbol{x})$ for all $\boldsymbol{x} \in \mathcal{S}$. Hence, if we have a neural network which can approximate $g : [-M, M]^d \to \mathbb{R}^p$, then we can compose it with a neural network which implements the JL embedding $\boldsymbol{A}$ to obtain a neural network which approximates $f : \mathcal{S} \to \mathbb{R}^p$. By pairing JL embedding existence results along with results on approximation of Lipschitz functions by neural networks, we obtain results which bound the complexity required for a neural network to approximate Lipschitz functions on high dimensional sets.

We now state our main theorem.

**Theorem 1.** *Let $d < D$ be positive integers, and let $L, M > 0$ and $\rho \in (0,1)$ be constants. Let $\mathcal{S} \subset \mathbb{R}^D$ be a bounded subset for which there exists a $\rho$-JL embedding $\boldsymbol{A} \in \mathbb{R}^{d \times D}$ of $\mathcal{S}$ into $[-M, M]^d$.*

*a) Suppose that any $\frac{L}{1-\rho}$-Lipschitz function $g : [-M, M]^d \to \mathbb{R}^p$ can be $\epsilon$-approximated by a feedforward neural network with at most $\mathcal{N}$ nodes, $\mathcal{E}$ edges, and $\mathcal{L}$ layers. Then, any $L$-Lipschitz function $f : \mathcal{S} \to \mathbb{R}^p$ can be $\epsilon$-approximated by a feedforward neural network with at most $\mathcal{N} + D$ nodes, $\mathcal{E} + Dd$ edges, and $\mathcal{L} + 1$ layers.*

*b) Furthermore, if there exists a single feedforward neural network architecture with at most $\mathcal{N}$ nodes, $\mathcal{E}$ edges, and $\mathcal{L}$ layers that can $\epsilon$-approximate any $\frac{L}{1-\rho}$-Lipschitz function $g : [-M, M]^d \to \mathbb{R}^p$, then there also exists another feedforward neural network architecture with at most $\mathcal{N} + D$ nodes, $\mathcal{E} + Dd$ edges, and $\mathcal{L} + 1$ layers that can $\epsilon$-approximate any $L$-Lipschitz function $f : \mathcal{S} \to \mathbb{R}^p$.*

*c) Suppose that the $\rho$-JL embedding is of the form $\boldsymbol{A} = \boldsymbol{M}\boldsymbol{D}$, where $\boldsymbol{M}$ is a partial circulant matrix, and $\boldsymbol{D}$ is a diagonal matrix with $\pm 1$ on its diagonal. Also, suppose that any $\frac{L}{1-\rho}$-Lipschitz*

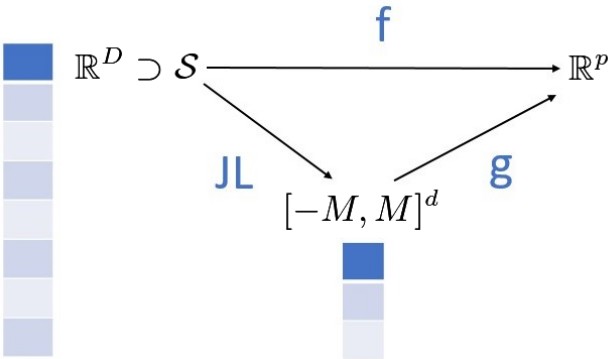

Figure 1: If there exists a $\rho$-JL embedding of $\mathcal{S} \subset \mathbb{R}^D$ into $[-M, M]^d$, then we can write the target function $f : \mathcal{S} \to \mathbb{R}^p$ as $f = g \circ JL$ where $g : [-M, M]^d \to \mathbb{R}^p$. So, we can then construct a neural network approximation of $f$ by using a neural network approximation of $g$ and adding a layer to implement the JL embedding.

*function $g : [-M, M]^d \to \mathbb{R}^p$ can be $\epsilon$-approximated by a convolutional neural network with at most $\mathcal{N}$ nodes, $\mathcal{P}$ parameters, and $\mathcal{L}$ layers. Then, any $L$-Lipschitz function $f : \mathcal{S} \to \mathbb{R}^p$ can be $\epsilon$-approximated by a feedforward neural network with at most $\mathcal{N} + 3D$ nodes, $\mathcal{P} + 2D + d$ parameters, and $\mathcal{L} + 4$ layers.*

*d) Furthermore, if there exists a single convolutional neural network architecture with at most $\mathcal{N}$ nodes, $\mathcal{P}$ parameters, and $\mathcal{L}$ layers that can $\epsilon$-approximate any $\frac{L}{1-\rho}$-Lipschitz function $g : [-M, M]^d \to \mathbb{R}^p$, then there also exists another convolutional neural network architecture with at most $\mathcal{N} + 2D$ nodes, $\mathcal{P} + 2Dd$ parameters, and $\mathcal{L} + 3$ layers that can $\epsilon$-approximate any $L$-Lipschitz function $f : \mathcal{S} \to \mathbb{R}^p$.*

*Remark* 1. The theorem ensures that the network size for approximating $f$ grows exponentially with the compressed dimension $d$ instead of growing exponentially with the input dimension $D$. The task now reduces to making the compressed dimension $d$ as small as possible while still ensuring that a $\rho$-JL embedding of $\mathcal{S}$ into $[-M, M]^d$ exists.

*Remark* 2. The theorem is quite general as parts a and b are not restricted to any particular type of network or activation function. In Section 3.3, we provide two corollaries of Theorem 1 that establish the expressive power of the feedforward and convolutional neural networks.

*Remark* 3. If an inverse operator is Lipschitz continuous and there exists a $\rho$-JL embedding of the set of possible observations $\mathcal{S}$ into $d$ dimensions, then the theorem gives us a bound on the complexity of a neural network architecture required to approximate the inverse operator.

## 3.1 JL EMBEDDINGS, AND COVERING NUMBERS AND GAUSSIAN WIDTH

As the existence of the JL map is a critical assumption of our theorem, in this section, we discuss the sufficient conditions for this assumption to hold. In addition, we also care about the structures of the JL maps, as they will end up being the first layer of the final neural network. For example, if the neural network is of convolution type, we need to make sure that a circulant JL matrix exists.

**Existence of $\rho$-JL maps**: It is well-known that for finite sets $\mathcal{S}$, the existence of a $\rho$-JL embedding can be guaranteed by the Johnson-Lindenstrauss Lemma. For sets $\mathcal{S}$ with infinite cardinally, the Johnson-Lindenstrauss lemma cannot be directly used. In the following proposition, we extend the Johnson-Lindenstrauss lemma from a finite set of $n$ points to a general set $\mathcal{S}$.

**Proposition 1.** *Let $\rho \in (0, 1)$. For $\mathcal{S} \subseteq \mathbb{R}^D$, define*

$$U_{\mathcal{S}} := \overline{\left\{ \frac{\boldsymbol{x} - \boldsymbol{x}'}{\|\boldsymbol{x} - \boldsymbol{x}'\|_2} \; : \; \boldsymbol{x}, \boldsymbol{x}' \in \mathcal{S} \; s.t. \; \boldsymbol{x} \neq \boldsymbol{x}' \right\}}$$

*to be the closure of the set of unit secants of $\mathcal{S}$, and $\mathcal{N}(U_{\mathcal{S}}, \|\cdot\|_2, \delta)$ to be the covering number of $U_{\mathcal{S}}$ with $\delta$-balls. Then, there exists a set $\mathcal{S}_1$ with $|\mathcal{S}_1| = 2\mathcal{N}(U_{\mathcal{S}}, \|\cdot\|_2, \delta)$ points such that if a matrix $\boldsymbol{A} \in \mathbb{R}^{d \times D}$ is a $\rho$-JL embedding of $\mathcal{S}_1$, then $\boldsymbol{A}$ is also a $(\rho + 2\|\boldsymbol{A}\|\delta)$-JL embedding of $\mathcal{S}$.*

The proposition guarantees that whenever we have a JL-map for finite sets, we can extend it to a JL-map for infinite sets with similar level of complexity measured in terms of the covering numbers. There are many known JL-maps for finite sets that we can extend from, including sub-Gaussian matrix (Matoušek, 2008), Gaussian circulant matrices with random sign flip (Cheng & Zhang, 2014), etc. We present some of the related results here.

**Proposition 2** ((Matoušek, 2008)). *Let $\boldsymbol{x}_1, \ldots, \boldsymbol{x}_n \in \mathbb{R}^D$. Let $\rho \in (0, \frac{1}{2})$ and $\beta \in (0, 1)$. Let $\boldsymbol{A} \in \mathbb{R}^{d \times D}$ be a random matrix whose entries are i.i.d. from a subgaussian distribution with mean 0 and variance 1. Then, there exists a constant $C > 0$ depending only on the subgaussian distribution such that if $d \geq C\rho^{-2} \log \frac{n}{\beta}$, then $\frac{1}{\sqrt{d}}\boldsymbol{A}$ will be a $\rho$-JL embedding of $\{\boldsymbol{x}_1, \ldots, \boldsymbol{x}_n\}$ with probability at least $1 - \beta$.*

**Proposition 3** (Corollary 1.3 in (Cheng & Zhang, 2014)). *Let $\boldsymbol{x}_1, \ldots, \boldsymbol{x}_n \in \mathbb{R}^D$. Let $\rho \in (0, \frac{1}{2})$, and let $d = O(\rho^{-2} \log^{1+\alpha} n)$ for some $\alpha > 0$. Let $\boldsymbol{A} = \frac{1}{\sqrt{d}}\boldsymbol{M}\boldsymbol{D}$ where $\boldsymbol{M} \in \mathbb{R}^{d \times D}$ is a random Gaussian circulant matrix and $\boldsymbol{D} \in \mathbb{R}^{D \times D}$ is a random Rademacher diagonal matrix. Then, with probability at least $\frac{2}{3}\left(1 - (D + d)e^{-\log^\alpha n}\right)$, $\boldsymbol{A}$ is a $\rho$-JL embedding of $\{\boldsymbol{x}_1, \ldots, \boldsymbol{x}_n\}$.*

Note that the $\alpha$ in the proposition can be set to be any positive number making the probability of failure less than 1.

Combining the results of Propositions 2 and 3 with Proposition 1, we have the following existence result for the JL map of an arbitrary set $\mathcal{S}$,

**Proposition 4.** *Let $\rho \in (0, 1)$ be a constant. For $\mathcal{S} \subseteq \mathbb{R}^D$, let $\mathcal{N}(U_{\mathcal{S}}, \|\cdot\|_2, \delta)$ to be the covering number with $\delta$-balls of the unit secant $U_{\mathcal{S}}$ of $\mathcal{S}$ defined in Proposition 1. Then*

*a) If $D \geq d \gtrsim \rho^{-2} \log \mathcal{N}(U_{\mathcal{S}}, \|\cdot\|_2, \frac{\rho}{4\sqrt{3D}})$, then there exists a matrix $\boldsymbol{A} \in \mathbb{R}^{d \times D}$ which is a $\rho$-JL embedding of $\mathcal{S}$.*

*b) If $D \geq d \gtrsim \rho^{-2} \log(4D + 4d) \log \mathcal{N}(U_{\mathcal{S}}, \|\cdot\|_2, \frac{\rho}{4\sqrt{3D}})$, then there exists a matrix $\boldsymbol{A} \in \mathbb{R}^{d \times D}$ in the form of $\boldsymbol{M}\boldsymbol{D}$ and of size $d \times D$ that works as $\rho$-JL map for $\mathcal{S}$, where $\boldsymbol{M}$ is a partial circulant matrix and $\boldsymbol{D}$ is a diagonal matrix with $\pm 1$ on its diagonal.*

The above proposition characterizes the compressibility of a set $\mathcal{S}$ by a JL-mapping terms of the covering number. Alternatively, one can also characterize it using the Gaussian width. For example, in (Iwen *et al.*, accepted. (See Arxiv 2110.04193)) it is shown using methods from (Vershynin, 2018) that if the set of unit secants of $\mathcal{S}$ has a low Gaussian width, then with high probability a subgaussian random matrix with provide a low-distortion linear embedding, and the dimension $d$ required scales quadratically with the Gaussian width of the set of unit secants of $\mathcal{S}$.

**Proposition 5** (Corollary 2.1 in (Iwen *et al.*, accepted. (See Arxiv 2110.04193))). *Let $\rho, \beta \in (0, 1)$ be constants. Let $\boldsymbol{A} \in \mathbb{R}^{d \times D}$ be a matrix whose rows $\boldsymbol{a}_1^T, \ldots, \boldsymbol{a}_d^T$ are independent, isotropic ($\mathbb{E}[\boldsymbol{a}_i\boldsymbol{a}_i^T] = \mathbf{I}$), and subgaussian random vectors. Let $\mathcal{S} \subset \mathbb{R}^D$, and Let*

$$\omega(U_{\mathcal{S}}) := \mathbb{E} \sup_{\boldsymbol{u} \in U_{\mathcal{S}}} \langle \boldsymbol{u}, \boldsymbol{z} \rangle, \quad \boldsymbol{z} \sim Normal(0, \mathbf{I})$$

*to be the Gaussian width of $U_{\mathcal{S}}$. Then, there exists a constant $C > 0$ depending only on the distribution of the rows of $\boldsymbol{A}$ such that if*

$$d \geq \frac{C}{\rho^2}\left(\omega(U_{\mathcal{S}}) + \sqrt{\log \frac{2}{\beta}}\right)^2,$$

*then $\frac{1}{\sqrt{d}}\boldsymbol{A}$ is a $\rho$-JL embedding of $\mathcal{S}$ with probability at least $1 - \beta$.*

In practice, one can use either the Gaussian width (Proposition 5) or the covering number (Proposition 4) to compute the lower bound of $d$, whichever is more convenient for a specific application.

## 3.2 Universal approximator neural networks for Lipschitz functions on $d$-dimensional cubes

In Theorem 1, we showed that with the help of JL, approximation rate of neural networks for functions defined on an arbitrary set $\mathcal{S}$ can be derived from their approximation rates for functions

defined on the cube $[-M, M]^d$. In this section, we review known results for the later, so that they can be used in combination of Theorem 1 to provide useful approximation results for network applications to various inverse problems. Specifically, we review two types of universal approximators for functions defined on the cube $[-M, M]^d$. One is the Feedforward ReLU network and the other is the Resnet type convolution neural network.

**Feedforward ReLU network:** The fully connected feedforward neural network with ReLU activation is known to be a universal approximator of any Lipschitz function on the box $[-M, M]^d$. Moreover, for such networks, the non-asymptotic approximation error has also been established, allowing us to get an estimate of the network size. The proposition below is a variant of Proposition 1 in (Yarotsky, 2018), and the proof uses an approximating function that uses the same ideas as in (Yarotsky, 2018).

**Proposition 6.** *Given constants $L, M, \epsilon > 0$ and positive integers $d$ and $p$, there exists a ReLU NN architecture with at most*

$$(p+C_1) \left(2 \left\lceil \frac{LM\sqrt{d}}{\epsilon} \right\rceil + 1\right)^d \text{ edges, } C_2 \left(2 \left\lceil \frac{LM\sqrt{d}}{\epsilon} \right\rceil + 1\right)^d + p \text{ nodes, and } \lceil \log_2(d+1) \rceil + 2 \text{ layers}$$

*that can $\epsilon$-approximate any $L$-Lipschitz function $g : [-M, M]^d \to \mathbb{R}^p$. Here, $C_1, C_2 > 0$ are universal constants. Also for each edge of the ReLU NN, the corresponding weight is either independent of $g$, or is of the form $g_i(\boldsymbol{x})$ for some fixed $\boldsymbol{x} \in [-M, M]^d$ and coordinate $i = 1, \ldots, p$.*

**Convolutional Neural Network:** As many successful network applications on inverse problems results from the use of filters in the CNN architectures (Jin *et al.*, 2017), we are particularly interested in the expressive power of CNN in approximating the Lipschitz functions. Currently known non-asymptotic results for CNN includes (Zhou, 2020; Petersen & Voigtlaender, 2020; Yarotsky, 2022), but they are established under stricter assumptions than merely Lipschitz continuous. On the other hand, the ResNet-based CNN with the following architecture has been shown to possess good convergence rate.

$$CNN_\theta^\sigma := FC_{W,b} \circ (\text{Conv}_{\omega_\mathbf{M}, \mathbf{b_M}}^\sigma + \text{id}) \circ \cdots \circ (\text{Conv}_{\omega_1, \mathbf{b_1}}^\sigma + \text{id}) \circ P \tag{1}$$

where $\sigma$ is the activation function, each $\text{Conv}_{\omega_\mathbf{m}, \mathbf{b_m}}$ is an convolution layer with $L_m$ filters $\omega_m^1, ..., \omega_m^{L_m}$ stored in $\omega_\mathbf{M}$ and $L_m$ bias $b_m^1, ..., b_m^{L_m}$ stored in $\mathbf{b_m}$. The addition by the identity map, $\text{Conv}_{\omega_\mathbf{M}, \mathbf{b_M}}^\sigma + \text{id}$, makes it a residual block. $FC_{W,b}$ represents a fully connected layer appended to the final layer of the network. We see that the ResNet-based CNN is essentially a normal CNN with skip connections.

The following asymptotic result is proved in (Oono & Suzuki, 2019). We note that the authors of (Oono & Suzuki, 2019) proved a more general result for $\beta$-Hölder functions, but we state it for Lipschitz functions, i.e., $\beta = 1$.

**Proposition 7** (Corollary 4 from (Oono & Suzuki, 2019)). *Let $f : [-1, 1]^d \to \mathbb{R}$ be a Lipschitz function. Then, for any $K \in \{2, ..., d\}$, there exists a CNN $f^{(CNN)}$ with $O(N)$ residual blocks, each of which has depth $O(\log N)$ and $O(1)$ channels, and whose filter size is at most $K$, such that $\|f - f^{(CNN)}\|_\infty \leq \widetilde{O}(N^{-1/d})$.*

### 3.3 Main results

We can now combine Propositions 4, 5, 6 and 7 with our Theorem 1 to obtain theorems bounding the required complexity of a feed-forward/convolutional neural network that can $\epsilon$-approximate any $L$-Lipschitz function on arbitrary sets $\mathcal{S} \subset \mathbb{R}^D$ for which a $\rho$-JL embedding into $[-M, M]^d$ exists.

**Theorem 2.** *Let $d < D$ be positive integers, and let $L > 0$ and $\rho \in (0, 1)$ be constants. Let $\mathcal{S} \subset \mathbb{R}^D$ be a bounded set and $U_\mathcal{S}$ be its set of unit secants. Suppose that*

$$d \gtrsim \min \left\{ \rho^{-2} \log \mathcal{N}(U_\mathcal{S}, \| \cdot \|_2, \tfrac{\rho}{4\sqrt{3D}}), \ \rho^{-2} \left(\omega(U_\mathcal{S})\right)^2 \right\},$$

*where $\mathcal{N}(U_\mathcal{S}, \| \cdot \|_2, \tfrac{\rho}{4\sqrt{3D}})$ is the covering number and $\omega(U_\mathcal{S})$ is the Gaussian width of $U_\mathcal{S}$. Then, there exists a ReLU neural network architecture with at most*

$$(p + C_1) \left(2 \left\lceil \frac{LM\sqrt{d}}{(1-\rho)\epsilon} \right\rceil + 1\right)^d + Dd \text{ edges,}$$

$$C_2 \left( 2 \left\lceil \frac{LM\sqrt{d}}{(1-\rho)\epsilon} \right\rceil + 1 \right)^d + p + D \ nodes,$$

$$and \ \lceil \log_2(d+1) \rceil + 3 \ layers$$

*that can $\epsilon$-approximate any L-Lipschitz function $f : \mathcal{S} \to \mathbb{R}^p$, where $M = \sup_{\boldsymbol{x} \in \mathcal{S}} \|\boldsymbol{Ax}\|_\infty$.*

Our Theorem 3 is a variant on our Theorem 2, but for convolutional neural networks.

**Theorem 3.** *Let $d < D$ be positive integers, and let $L > 0$ and $\rho \in (0,1)$ be constants. Let $\mathcal{S} \subset \mathbb{R}^D$ be a bounded set and $U_\mathcal{S}$ be its set of unit secants. Suppose that*

$$d \gtrsim \rho^{-2} \log(4D + 4d) \log \mathcal{N}(U_\mathcal{S}, \| \cdot \|_2, \tfrac{\rho}{4\sqrt{3D}}).$$

*Then, for any L-Lipschitz function $f : \mathcal{S} \to \mathbb{R}^p$, there exists a ResNet type CNN $f^{(CNN)}$ in the form of (1) with $O(N)$ residual blocks, each of which has a depth $O(\log N)$ and $O(1)$ channels, and whose filter size is at most $K$ such that $\|f - f^{(CNN)}\|_\infty \leq \widetilde{O}(N^{-1/d})$.*

## 4 APPLICATIONS TO INVERSE PROBLEMS

Now we focus on inverse problems and demonstrate how the main theorems can be used to provide a reasonable estimate of the size of the neural networks needed to solve some classical inverse problems in signal processing. The problems we consider here are sparse recovery, blind deconvolution, and matrix completion.

In all the inverse problems, we want to recover some signal $x \in \mathcal{S}$ from its forward measurement $y = F(x)$, where the forward map $F$ is assumed to be known. The minimal assumption we have to impose on $F$ is the invertibility.

**Assumption 1** (invertibility of the forward map): Let $\mathcal{S}$ be the domain of the forward map $F$, and $\mathcal{Y} = F(\mathcal{S})$ be the range. Assume that the inverse operator $F^{-1} : \mathcal{Y} \to \mathcal{S}$ exists and is Lipschitz continuous with constant $L$,

$$\|F^{-1}(y_1) - F^{-1}(y_2)\| \leq L\|y_1 - y_2\|, \quad \text{for all} \quad y_1, y_2 \in \mathcal{Y}.$$

For any inverse problems satisfying Assumption 1, Theorem 1, 2 and 3 provide ways to estimate the size of the universal approximator networks for the inverse map. When applying the theorems to each problem, we need to estimate the covering number of $U_\mathcal{Y}$ first.

Depending on the problem, one may estimate the covering number either numerically or theoretically. If the domain $\mathcal{Y}$ of the inverse map is irregular and discrete, then it may be easier to compute the covering number numerically. If the domain has a nice mathematical structure, then we may be able to estimate it theoretically. Below are three examples of the theoretical estimation. From them, we see that it is quite common for inverse problems to have a small intrinsic complexity, with which Theorems 2 and 3 can significantly reduce the required size of the network from the previously known results.

We emphasize that the covering number that the theorems use is the one of the unit secant of $\mathcal{Y}$, which can be much larger than the covering number of $\mathcal{Y}$ itself.

**Sparse recovery:** Sparsity is now one of the most commonly used priors in inverse problems as signals in many real applications possess certain level sparsity in some appropriate domain. For simplicity, we consider the strictly sparse signals. Let $\Sigma_s^N$ be the set of $s-$sparse vectors of length $N$. Assume a sparse vector is measured linearly $y = Ax \equiv F(x)$, the inverse problem amounts to recovering $x$ from $y$. Now that we want to use a network to approximate the inverse map $F^{-1} : A\Sigma_s^N \equiv \mathcal{Y} \in y \to x \in \Sigma_s^N$, and estimate the size of the network through the theorems, we need to estimate the covering number of the unit secant $U_{A\Sigma_s^N}$.

**Proposition 8.** *Let $U_{A\Sigma_s^N}$ denote the set of unit secants of $A\Sigma_s^N$. Then, we have*

$$\log \mathcal{N}(U_{A\Sigma_s^N}, \| \cdot \|, \delta) \lesssim s \log \frac{N}{\delta}.$$

*Proof.* By definition, the unit secant of $\mathcal{Y} = A\Sigma_s^N$ is defined as

$$U_{\mathcal{Y}} = \left\{ \frac{y_1 - y_2}{\|y_1 - y_2\|}, \quad y_1, y_2 \in A\Sigma_s^N \right\}$$

which contains all unit vectors that are linear combinations of $2s$ columns of $A$. Let $T$ with $|T| = 2s$ be a fixed support set, the covering number of $\text{span}(A_T) \cap \mathbb{S}^{m-1}$ is $(\frac{3}{\delta})^{2s}$, so the covering number of $U_{\mathcal{Y}}$ is at most $N^s(\frac{3}{\delta})^{2s}$.

$\square$

If the inverse of $F$ exists, such as in the case when $A$ is a Restricted-Isometry-Property matrix, then by Theorem 2 and 3, there exist neural networks of fully connected type or of CNN type with $O(\epsilon^{-s \log N})$ number of weights, that can do the sparse recovery up to an error of $\epsilon$.

**Blind deconvolution:** Blind deconvolution concerns the recovery of a signal $x$ from its blurry measurements

$$y = k \otimes x \tag{2}$$

when the kernel $k$ is also unknown. Here $\otimes$ denotes the convolution operation.

Blind-deconvolution is an ill-posed problem due to the existence of a scaling ambiguity between $x$ and $k$, namely, if $(k, x)$ is a solution, then $(\alpha k, \frac{1}{\alpha} x)$ with $\alpha \neq 0$ is also a solution. To resolve this issue, we focus on recovering the outer product $xk^T$, where $x$ and $k$ here are both columns vectors. The recovery of the outer product $xk^T$ from the convolution $y = k \otimes x$ can be well-posed in various settings (Lee *et al.*, 2015; Ahmed *et al.*, 2013). For example, (Ahmed *et al.*, 2013) showed that if we assume $x = \Phi u$ and $k = \Psi v$, where $\Phi \in \mathbb{R}^{N,n} (n < N)$ is i.i.d. Gaussian matrix and $\Psi \in \mathbb{R}^{N,m} (m < N)$ is a matrix of small coherence, then for large enough $N$, the outer-product $xk^T$ can be stably recovered from $y$ in the following sense. For any two signal-kernel pairs $(x, k)$, $(\tilde{x}, \tilde{k})$ and their corresponding convolutions $y, \tilde{y}$, we have

$$\|xk^T - \tilde{x}\tilde{k}^T\| \leq L\|y - \tilde{y}\| \tag{3}$$

with some $L$. When using a neural network to approximate the inverse map by $F^{-1} : y \to xk^T$, we need to estimate the covering number of the unit secant cone of $\mathcal{Y} = \{y = x \otimes k, x \in \Phi\Sigma_s^N, k \in \text{span}\Psi\}$, which is done in the following proposition.

**Proposition 9.** *Suppose the inverse map $F^{-1} : y \to xk^T$ is Lipschitz continuous with Lipschitz constant $L$, then for $\mathcal{Y} = \{y = x \otimes k, x \in \text{span}(\Phi), k \in \text{span}\Psi\}$, the logarithm of the covering number of the set of unit secants of $\mathcal{Y}$ is bounded by*

$$\log \mathcal{N}(U_{\mathcal{Y}}, \|\cdot\|_2, \delta) \lesssim \max\{m, n\} \log \frac{3L}{\delta}.$$

Combining this proposition with Theorem 2 and 3, we obtain that there exist neural networks of full connected type or of CNN type having about $O(\epsilon^{-\max\{m,n\} \log(L(n+m))})$ number of weights, that can solve the blind-deconvolution problem up to an error of $\epsilon$.

**Matrix completion:** Matrix Completion is a central task in machine learning where we want to recover a matrix from its partially observed entries. It arises from a number of applications including image super resolution (Shi *et al.*, 2013; Cao *et al.*, 2014), image/video denoising (Ji *et al.*, 2010), recommender systems (Zheng *et al.*, 2016; Monti *et al.*, 2017), and gene-expression prediction (Kapur *et al.*, 2016), etc.. Recently neural network models have achieved state-of-the-art performance (Zheng *et al.*, 2016; Monti *et al.*, 2017; Dziugaite & Roy, 2015; He *et al.*, 2017), but a general existence result in the non-asymptotic regime is still missing.

In this setting, the measurements $Y = P_\Omega X$ consists of a set of observed entries of the unknown low-rank matrix $X$, where $\Omega$ is the index set of the observed entries and $P_\Omega$ is the mask that sets all but entries in $\Omega$ to 0. Assuming $M_r^{n,m}$ is the set of $n \times m$ matrices with rank at most $r$ and $X \in M_r^{n,m}$. If the mask is random, and the left and right eigenvectors $U, V$ of $X$ are incoherent, in the sense that

$$\max_{1 \leq i \leq n} \|U^T e_i\| \leq \sqrt{\frac{\mu_0 r}{n}}, \quad \max_{1 \leq i \leq m} \|V^T e_i\| \leq \sqrt{\frac{\mu_0 r}{m}}, \tag{4}$$

$$\max_{1 \leq i \leq n, 1 \leq j \leq m} \|(UV^T)_{i,j}\| \leq \sqrt{\frac{\mu_1 r}{nm}}$$

then it is known (e.g. (Candes & Plan, 2010)) that provided that the number of observations

$$|\Omega| \gtrsim \mu_0 r \max\{m, n\} \log^2 \max\{m, n\},$$

then with overwhelming probability, the inverse map $F^{-1} : Y \to X$ exists and is Lipschitz continuous. Let us denote the set of low-rank matrices satisfying (4) to be $\mathcal{C}$. To estimate the complexity of the inverse map, we compute the covering number of $U_{\mathcal{Y}}$ for $\mathcal{Y} = \{Y = P_\Omega X : X \in M_r^{m,n} \cap \mathcal{C}\}$.

**Proposition 10.** *Suppose the mask is chosen so that the inverse map $F^{-1} : Y = P_\Omega X \to X$ is Lipschitz continuous with Lipschitz constant $L$, then for $\mathcal{Y} = \{P_\Omega X : X \in M_r^{m,n} \cap \mathcal{C}\}$, the logarithm of the covering number of the set of unit secants of $\mathcal{Y}$ is bounded by*

$$\log \mathcal{N}(U_{\mathcal{Y}}, \|\cdot\|_2, \delta) \lesssim r(m + n) \log\left(\frac{L}{\delta}\right).$$

Combining this proposition with Theorem 2 and 3, we obtain that there exist neural networks of full connected type or of CNN type having about $O(\epsilon^{-r(m+n)\log(Lnm)})$ number of weights, that can solve the blind-deconvolution problem up to an error of $\epsilon$.

## 5    CONCLUSION AND DISCUSSION

The main message of this paper is that when neural networks are used to approximate Lipschitz continuous functions, the size of the network only needs to grow exponentially with respect to the intrinsic complexity of the input set measured using either the Gaussian width or the covering numbers. Therefore, it is more optimistic than the previous estimate that requires the size of the network to grow exponentially with respect to the input dimension.

Similar results were derived previously in (Chen *et al.*, 2019) in a more restrictive setting, namely, the input set is assumed to be close to a smooth manifold with a small curvature, and the network type is restricted to the feedforward ReLU networks. In this paper, by utilizing the JL map, we are able to state the result in a very general setting, that does not pose any structural requirement on the inputs set other than that they have a small complexity. In addition, our result holds for many different types of networks – although we only stated it for feedfoward neural networks and the ResNet type of convolutional neural networks, the same idea naturally applies to other types of networks as long as an associated JL-map exists.

The estimate we provided for the network size ultimately depends on the complexity of the input set, measured by either the covering number or the Gaussian width of its set of unit secants. The computation of these quantities varies case by case, and in some cases might be rather difficult. This is a possible limitation of the proposed method. Because if the estimation of the input complexity is not tight enough, we may again get a pessimistic bound. Having said that, for most of the classical inverse problems, the covering number and the Gaussian width are not too difficult to calculate. As we demonstrated in Section 4, there are many known properties of them that one can use to facilitate the calculation. And when a training dataset is given, one can even compute the covering number numerically with off-the-shelf algorithms.

Finally, although the applications of neural networks to inverse problems are seeing its success. There are much more failed attempts with unclear reasons. One common explanation is that the size of the network in use is not large enough for the targeted applications. Since inverse problems models usually have a much higher intrinsic dimensionality than say image classification models, the required network sizes might also be much larger. The classical universal approximation theorems only guarantees small errors when the network size approaches infinity, therefore is not very helpful in the non-asymptotic regime where we have to choose the network size, which is now known to be critical to good performances. We hope the presented result can give more insight on this matter.

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
