# OpenReview forum: "Neural Network Approximation of Lipschitz Functions in High Dimensions with Applications to Inverse Problems"
_ICLR.cc/2023/Conference — Submitted to ICLR 2023_

### Official Review · Reviewer_RMbx · 2022-10-24

**Confidence:** 4
**Clarity, Quality, Novelty And Reproducibility:** The paper is well written and the res…
**Correctness:** 3
**Technical Novelty And Significance:** 3
**Empirical Novelty And Significance:** 3
**Recommendation:** 6

**Strength And Weaknesses:**

I think the main limitation of the theory is about that the model (y=F(x)) considered in this paper has no noise term, which is not very common in the literature of inverse problems.
- If the results could be generalized to this case, the contribution would be stronger. It not, a discussion regarding this point is welcome in the conclusion.
- Do you have an estimation of the d in practice, so as to justify the theory is relevant to practice?

**Summary Of The Paper:**

This paper studies how to approximate Lipschitz functions in high dimensions for signals with low-dimensional structures. By assuming the existence of a linear Johnson-Linderstrauss embedding on the signals, the appropriation bounds of Lipschitz functions based on neural networks are given. It allows for better explanation of the empirical success of deep learning models in inverse problems.

**Summary Of The Review:**

Revision: I tend to keep my score it is still not so clear how the results could be extended to noisy cases (without modifying Assumption 1). Also the application to inverse problems remain theoretical, therefore I think the impact of the paper is limited.

---

> ### Author Response · Authors · 2022-11-16
> **Response to Reviewer RMbx**
>
> Indeed, it is of interest to note how robust our neural network approximation is to input noise. Suppose we observe $y = F(x)+\eta$ for some noise $\eta \in \mathbb{R}^D$ and aim to recover $x \in \mathbb{R}^p$ using a neural network that approximates the inverse map.
>
> To restate a few details of our paper for the inverse problem setting, if the inverse map $F^{-1} : \mathcal{S} \to \mathbb{R}^p$ is $L$-Lipschitz, and $\mathbf{A} \in \mathbb{R}^{d \times D}$ is a $\rho$-JL embedding of $\mathcal{S}$ into $[-M,M]^d$, then there exists a $\tfrac{L}{1-\rho}$-Lipschitz function $g : [-M,M]^d \to \mathbb{R}^p$ such that $g(\mathbf{A} y) = F^{-1}(y)$ for all $y \in \mathcal{S}$. Let $g_{\text{NN}} : [-M,M]^d \to \mathbb{R}^p$ be a neural network that can $\epsilon$-approximate $g$, and let $F_{\text{NN}}^{-1} = g_{\text{NN}} \circ \mathbf{A}$.
>
> Then, as long as the noise $\eta$ is small enough to keep $\mathbf{A}(F(x)+\eta) \in [-M,M]^d$, we can bound the approximation error in using the neural network $F_{\text{NN}}^{-1}$ to recover $x$ as follows:
>
> $\\|F_{\text{NN}}^{-1}(F(x)+\eta)-x\\|$
>
> $= \\|F_{\text{NN}}^{-1}(F(x)+\eta)-F^{-1}(F(x))\\|$
>
> $= \\|g_{\text{NN}}(\mathbf{A}(F(x)+\eta))-g(\mathbf{A}(F(x)))\\|$     (Applying the definitions $F_{\text{NN}}^{-1} = g_{\text{NN}} \circ \mathbf{A}$ and $g(\mathbf{A} \ \cdot) = F^{-1}(\cdot)$ on $\mathcal{S}$)
>
> $\le \\|g_{\text{NN}}(\mathbf{A}(F(x)+\eta))-g(\mathbf{A}(F(x)+\eta))\\|+\\|g(\mathbf{A}(F(x)+\eta))-g(\mathbf{A}(F(x)))\\|$ (triangle inequality)
>
> $\le \epsilon+\tfrac{L}{1-\rho}\\|\mathbf{A}\eta\\|,$
>
> where the last line follows from the fact that $g_{\text{NN}}$ is an $\epsilon$-approximation to $g$ on $[-M,M]^d$ and $g$ is $\tfrac{L}{1-\rho}$-Lipschitz.
>
> Typical JL maps from $D$-dimensions to $d$-dimensions satisfy $\\|\mathbf{A}\\| \lesssim \sqrt{D/d}$, which gives us a worst case bound of $\lesssim \epsilon + \tfrac{L}{1-\rho}\sqrt{\tfrac{D}{d}}\\|\eta\\|$. However, if $\eta \in \mathbb{R}^D$ has i.i.d. Gaussian entries, then it can be shown that with high probability $\\|\mathbf{A}\eta\\| \lesssim \\|\eta\\|$, which gives us the high probability bound of $\lesssim \epsilon + \tfrac{L}{1-\rho}\\|\eta\\|$. We will be sure to add this discussion on the impacts of noise to our paper when revising it. Thank you for suggesting that we add this discussion.
>
> -----
>
> The embedding dimension $d$ will depend on how large of a problem one is working with. In Section 4, we bound the logarithm of the covering number for the set of unit secants of the input spaces for some common inverse problems. Proposition 4 says that the embedding dimension $d$ needs to be a small constant factor larger than the logarithm of the covering number. In our revision, we will link these two facts more explicitly to state a bound on the embedding dimension $d$ in each of the inverse problems. Thank you for pointing out that we need to link these two facts more explicitly.

---

> ### Author Response · Authors · 2022-12-02
> **Updated Response to Reviewer RMbx**
>
> Thank you for reading our response. We have elaborated some of the steps for bounding the recovery error when using our NN to solve an noisy inverse problem. Could you please clarify if any of the steps above are unclear or if you meant something else regarding generalizing the results to the noisy case?
>
> -----
>
> Also, regarding the values of $d$ in practice, (Pope et. al. 2021) show that many image datasets used in practice (including CIFAR-100 and ImageNet) have intrinsic dimensions that are between $7$ and $43$ (see Table 1 in Pope et. al. 2021 for the datasets and estimated intrinsic dimensions).
>
> -----
>
> Update: We recently performed numerical experiments where we trained a neural network to perform sparse recovery. The results show that compressing the measurements with a JL map prior to feeding it into the neural network can actually improve performance. See this link for details. https://anonymous.4open.science/r/ICLR2023_Paper3295-C885/SparseRecoveryExperiment.pdf
>
> -----
>
> Phillip Pope, Chen Zhu, Ahmed Abdelkader, Micah Goldblum, and Tom Goldstein. The intrinsic dimension of images and its impact on learning. arXiv:2104.08894, 2021.

---

### Official Review · Reviewer_ZXNm · 2022-10-24

**Confidence:** 3
**Correctness:** 3
**Technical Novelty And Significance:** 2
**Empirical Novelty And Significance:** Not applicable
**Recommendation:** 5

**Clarity, Quality, Novelty And Reproducibility:**

The paper is clear to follow, however seems to be lacking novelty in the key parts of the result.

**Strength And Weaknesses:**

[Strengths]

The results provided in the paper are technically correct, and easy to follow and the proofs are well presented.

The authors also provide conditions under which it is reasonable for a JL to exist.

The authors provide the estimates of size for solving some classical problems like sparse recovery and matrix completion.

[Weakness]

The key contribution of the paper is dependent upon the existence of a JL-map $A$. However it is unclear if such a map can exist in natural domains for which neural networks are used.

The proof techniques used to show epsilon-approximation in the intrinsic dimension, and the resulting size of the neural networks (in terms of depth, width and parameters) is very similar to previous works by Yarotsky (2018,2022) and indeed borrows a lot of details from the their proof. The key idea of the existence of a JL-Map $A$ does not seem like sufficiently novel.

**Summary Of The Paper:**

In this paper the authors answer why the size of a Lipschitz neural network in practice is much smaller than the theoretical bounds (which have an exponential in D dependence). The key idea that the authors provide is that number of parameters will vary exponential in the intrinsic dimension if there exists a matrix $A$ which projects a sample from higher dimension D to d, such that it also preserves the distance between the points.

Therefore, if there exists a neural network with a given width, depth and parameters (epsilon-dependent) that can epsilon-approximate a Lipschitz d, then it can also approximate a function in D with only few extra parameters and layers.

**Summary Of The Review:**

The techniques used to prove the main theorem seem to build heavily upon existing results, due to which I am currently leaning towards a weak-reject. I am open to having a discussion with the authors in this regard.

---

> ### Author Response · Authors · 2022-11-16
> **Response to Reviewer ZXNm**
>
> In Section 4, we consider some common inverse problems and derive bounds for the logarithm of the covering number of the set of unit secants of the input space. Proposition 4 says that the embedding dimension $d$ needs to be a small constant factor larger than the logarithm of the covering number. As such, this JL embedding does exist for some common inverse problems. In our revision, we will link these two facts more explicitly to state a bound on the embedding dimension $d$ in each of the inverse problems in addition to the bound on the logarithm of the covering number. Thank you for pointing out that we need to link these two facts more explicitly.
>
> -----
>
> In regards to the novelty of our work, while our proof borrows ideas from Yarotsky's previous works, so do other approximation results for neural networks. For example, in (Chen et. al. 2019), they cover the $d$-dimensional manifold with small charts $\bigcup_i U_i$. For each chart $U_i$, they construct a linear map $\phi_i$ (which is a projection onto the tangent space $T_{c_i}(\mathcal{M})$ at a point $c_i \in U_i$ followed by a rescaling) so that if $x \in U_i$ then $\phi_i(x) \in [0,1]^d$. They then use Yarotsky's results for constructing a neural network to approximate a $C^n$ function on $[0,1]^d$.
>
> (Chen et. al. 2019) also borrows an idea from (Yarotsky 2017) of forming a partition of unity, i.e., a collection of functions $\rho_i : \mathcal{M} \to [0,1]$ such that $\sum_{i}\rho_i \equiv 1$ on $\mathcal{M}$ and $\text{supp}(\rho_i) \subset U_i$, and then approximating $f \approx \sum_i \widehat{\times}(f_i,\rho_i)$ where $f_i \approx f$ on $U_i$ and $\widehat{\times}(\cdot , \cdot)$ is a ReLU neural network approximation of the multiplication operation. (Yarotsky 2017) uses this idea, but for the unit hypercube $[0,1]^d$.
>
> Our construction allows us to use the same linear map for the entire set $\mathcal{S} \subset \mathbb{R}^D$ instead of a different linear map for each local piece. This also makes combining the local approximations into a single neural network easier. While our proof simpler, it also allows us to obtain neural network approximation results on a wider variety of domains for which other techniques cannot. Furthermore, our idea can be used to extend universal approximation results for other types of neural networks from low-dimensional boxes to high-dimensional, low-complexity sets. See Theorem 3 as an example with ResNet type CNNs.
>
> -----
>
> Dmitry Yarotsky. Error bounds for approximations with deep relu networks. Neural Networks, 94:103–114, 2017.
>
> Minshuo Chen, Haoming Jiang, Wenjing Liao, and Tuo Zhao. 2019. Efficient approximation of deep relu networks for functions on low dimensional manifolds. Advances in neural information processing systems, 32

---

> > ### Comment · Reviewer_ZXNm · 2022-11-23
> > **Thank you for your response**
> >
> > Thank you for the rebuttal. I think the point about the existence of JL embedding existing for natural datasets still remain (I don’t think that the authors have made any changes to the manuscript regarding that). I still think that the construction borrows heavily from previous work and would like to keep my score for now.

---

> > > ### Author Response · Authors · 2022-12-02
> > > **Reply to Reviewer ZXNm**
> > >
> > > Thank you for reading our response. We apologize that we misinterpreted what you meant by "natural domains" in your review. In regards to JL embeddings for natural datasets, (Pope et. al. 2021) show that many image datasets used in practice (including CIFAR-100 and ImageNet) have intrinsic dimensions that are between $7$ and $43$ (see Table 1 in Pope et. al. 2021 for the datasets and estimated intrinsic dimensions). If these are not the kinds of natural datasets you are looking for, please let us know, and we'll gladly try to find other examples.
> > >
> > > -----
> > >
> > > Phillip Pope, Chen Zhu, Ahmed Abdelkader, Micah Goldblum, and Tom Goldstein. The intrinsic dimension of images and its impact on learning. arXiv:2104.08894, 2021.

---

### Official Review · Reviewer_Q6xo · 2022-10-26

**Confidence:** 4
**Correctness:** 4
**Technical Novelty And Significance:** 2
**Empirical Novelty And Significance:** Not applicable
**Recommendation:** 5

**Clarity, Quality, Novelty And Reproducibility:**

The paper is clearly written and easy to understand. (There are quite a few misprints and the prose could be streamlined, but that should be easy to fix.) As for originality, I have some doubts expressed in the previous box. The theme of rates that adapt to the intrinsic dimension of data is old and well-understood so I occasionally had the impression that the high-level idea is a bit oversold. My main objection is that I know of prior work that addresses the same problem and derives considerably more general results.

**Strength And Weaknesses:**

The theme of approximation rates that adapt to intrisic dimensionality is a familiar one in machine learning and signal processing. The current manuscript studies a natural construction where the data is first linearly projected into a low-dimensional Euclidean space. The injectivity and stability of this projection are guaranteed by the JL lemma and its extension to continuous sets provided in the manuscript.

In my opinion the main strength of the manuscript is that it focuses on a specific but still sufficiently general setting, and that it uses familiar tools to get a very explicit result that is natural and easy to understand. As far as I can tell, the mathematical arguments are sound, and the presentation is clear.

On the critical side, the novelty is in my opinion limited in view of existing work. The presented results seem to be a special case of

``Universal Approximation Theorems for Differentiable Geometric Deep Learning'', JMLR, Anastasis Kratsios, Léonie Papon; 23(196):1−73, 2022.

To see this, set

- $I = 1$
- $X = \text{the subset S of}~\mathbb{R}^D$
- $Y = R^p$
- $\varphi(x) := (1/2M) * (Sx - (M,\ldots,M))$; i.e. the JL-embeding S->R^d and the rescaling of the cube [-M,M]^d to the unit cube [0,1]^d
	note that this map is injective and continuous (since it is bi-Lipschitz) whence it satisfies Assumption 7
- $\rho$ be the identity on R^p
- Take the exponential maps to be about the origin; i.e.
- Exp_{R^d,0}(u) = u +0 =u
- Exp_{R^p,0}(u) = u +0 =u

In fact, the above paper derives sharper approximation rates that are dimension-free when $f \circ \varphi^{-1}$ admits a Whitney-type extension of $\varphi(S)$ (Corollary 45). Related arguments are used Theorem 3.3 in ``Non-Euclidean Universal Approximation'' by Kratsios and Bilokopytov; in particular that should allow to substitute any universal regressor.


The present work is also closely related to

- Deep Fried Convnets: https://openaccess.thecvf.com/content_iccv_2015/papers/Yang_Deep_Fried_Convnets_ICCV_2015_paper.pdf
- NN approximation w/ random projections: https://core.ac.uk/download/pdf/153400703.pdf
- The classic work on nonlinear learning with local coordinate coding: https://papers.nips.cc/paper/2009/file/2afe4567e1bf64d32a5527244d104cea-Paper.pdf
- Direct inference on compressive measurements: https://ieeexplore.ieee.org/document/7532691
- Sketching and neural networks: https://arxiv.org/pdf/1604.05753.pdf

### A few additional remarks

- You mention that a mild drawback of the work of Chen et al. is that the bounds depend on manifold curvature which is unknown or hard to estimate. But in your JL-based results the covering number of the secant set will surely depend on similar parameters of the data domain. How would you reliably estimate covering numbers or Gaussian widths from data?

- Your architecture first reduces dimension, a bit like an autoencoder, but standard architectures for inverse problems in imaging don't do that (one example is the U-Net). They first create many channels to expand dimensionality and then reduce it back to the image dimension. This demonstrably improves the expressivity. These architectures are very successful at addressing inverse problems, even at high resolutions. How does this fit your narrative? A related comment is that there are many situations where overparameterization is known to help (random kitchen sinks, most neural networks). In some cases we have theory. Again, it would be nice to hear how this combines with your narrative. Somehow I think that (admittedly vague) notions like inductive bias will deteriorate after compression.

- One thing that will certainly be problematic is that in many problems with convolutional structure a "good" architecture can opportunely leverage local information in the full-dimensional input (e.g., skip connections in the U-Net or the ResNet). A low-dimensional projection is likely to destroy this structure.

- I am slightly confused by the convolutional results (Theorem 3). Since there is no equivariance assumption on f, the output space of the network, R^p, should not be interpreted as an image? (assuming that we work with images). The result of Zhou, for example, seems to study a different model than the one used in inverse problems (where again in the case of images the channels and the output of a convolutional network like the U-net are image-like + equivariance plays an important role). Zhou as well as Oono & Suzuki seem to approximate a generic f \in C(Omega) and not leverage any image-related inductive bias of the domain or the range of f.

### Questions, suggestions, comments


- The covering numbers in Proposition 1 scale favorably when the set in question has some low dimensional structure (for example, it's an embedded manifold). It might be good to state this explicitly and write down some common scalings. Also, I assume there is a relation which such things as the box dimension and doubling constants.


- In compressive imaging the input to the reconstruction map is low dimensional to begin with and presumably cannot be compressed any further. Could you comment on the most natural classes of problems where your results apply?

- A philosophical comment: Your initial assumption is that the inverse operator is Lipschitz. I understand that you're looking at a problem which is already discretized, but many inverse operators in interesting inverse problems are not Lipschitz between function spaces on which they are usually defined. This is the case even for the Radon transform which smooths by half a derivative (in 2S) and thus has a Hölder inverse, though this can be "made Lipschitz" by changing norms. But in problems like electric impedance tomography, the inverse problem is only log stable for any reasonable norm (this is very bad). By the results of Bourgain and later others (e.g. Stefanov and Uhlman), once we go to finite dimension things do become Lipschitz but the Lipschitz constants will still reflect the fundamental instability of the continuous problem.


### A few misprints I caught


Introduction

- Nah et al. 2017 is a deblurring paper but it is cited in the context of low-does CT reconstruction; please double check all references

- Spurious ) after m < n in the second paragraph

- In paragraph 3, The inverse of the linear measurement map -> delete "inverse of the"

- Second sentence of paragraph 4: delete either alternatively or instead

Section 2: Related work

- The sentence starting with "Therefore, to study expressive power... " is broken
- Last paragraph on p2: Guassian -> Gaussian


Section 3: Main results

- The sentence "Since our theory is to be applied..." is broken
- The title of 3.1 has two ands




**Summary Of The Paper:**

The authors use the Johnson--Lindenstraus lemma to argue that a Lipschitz function over a suitable low-dimensional subset of a high-dimensional ambient space can be approximated by neural networks whose complexity mainly depends on the intrinsic dimensionality of the low-dimensional set, and only weakly on the ambient dimension. The authors extend the JL lemma to domains whose unit secant sets have controlled covering numbers or Gaussian width (effectively: low-dimensional domains). They give results for fully-connected and convolutional networks and apply them to two compressed-sensing-style problems.


**Summary Of The Review:**

This is a nice paper with a good, important idea, and a clear message. I think the presentation should include a careful comparison to earlier related results in the literature (some of which address the exact same problem and even make stronger statements). I very much appreciate the motivation via inverse problems but that also raises certain questions about inductive bias (arguably _the_ reason why the current architectures work so well across the board), and mangling this inductive bias by low-dimensional projections. But the bulk of the basis for my rating is the overlap of the result with the previous (more general) work of Kratsios and Papon, cited above. I think that the specific setting studied by the authors _is_ among the most interesting instances of this theory and warrants a specialized treatment, but I feel that this requires a substantial revision to properly reflect existing results.

---

> ### Author Response · Authors · 2022-11-16
> **Response to Reviewer Q6xo**
>
> While we have not had the time to read (Kratsios and Papon 2022) in extreme detail, we noticed that Remark 21 says that if the dataset $\mathbb{X}$ is $1$-efficient for $f$, then the network in Theorem 20 has depth of order roughly $O(m+m\epsilon^{-1/3})$ and needs $\approx O(m^2\epsilon^{-2/3})$ trainable parameters (where $m$ is the output dimension in that paper, which is $p$ in our paper). Corollary 45 also has similar bounds for the depth and number of trainable parameters as Theorem 20. While the network in that paper has a smaller number of trainable parameters, it also has a larger depth. We believe this is enough of a difference so that our result is not a special case of the result in that paper.
>
> -----
>
> Thank you for bringing (Kratsios and Papon 2022) and the other papers to our attention. We will definitely cite (Kratsios and Papon 2022) some of the other papers in our revision.
>
> -----
>
> Section 3 of (Kégl. 2002) gives an algorithm for estimating the covering number of a set. Also, the Gaussian width of the set of unit secants of a set of data points $U_X$, i.e. $\omega(U_X) = \mathbb{E} \sup_{u \in U_X} \langle g,u \rangle$ can be estimated via Monte Carlo integration. Of course, this will only serve as a lower bound since there is always chance we won't have samples from large areas of the manifold. We will add a short discussion on estimating the covering number and Gaussian width of a set from samples when revising our paper. Thank you for this suggestion.
>
> -----
>
> In Section 4, we show how the covering number scales for some common inverse problems. In our revision, we will add a sentence or two near Proposition 1 to highlight this and mention how the covering number scales for manifolds. Thank you for suggesting for us to write down some common scalings so that readers who are less familiar with the covering number can benefit.
>
> -----
>
> One inverse problem where our results naturally apply is blind deconvolution, i.e., we observe $y = k \circledast x \in \mathbb{R}^N$ where $k$ and $x$ come from known low-dimensional subspaces. In section 4, we show that the embedding dimension for the set of possible observations is a log-factor times the larger of the dimensions of the two subspaces. This can be much smaller than the dimension of the observed convolution $y$.
>
> -----
>
> Thank you very much for catching those misprints. We will be sure to correct them when revising this work.
>
> -----
>
> Balázs Kégl. Intrinsic dimension estimation using packing numbers. Advances in neural information processing systems, 15, 2002.

---

> > ### Comment · Reviewer_Q6xo · 2022-11-22
> > **thank you + keeping my score**
> >
> > Thank you for the response. Unfortunately many of my comments were not at all addressed. Even the remark about the (much more general) work of Kratsios and Papon does not offer any serious discussion. Finally, as far as I can tell, no updates were made in the manuscript. I therefore retain my initial score—I do not think that this manuscript is ready for publication in its current state.

---

> > > ### Author Response · Authors · 2022-12-02
> > > **Reply to Reviewer Q6xo**
> > >
> > > Thank you for reading our response. We apologize for not being able to respond to all of your comments earlier. We needed time to thoroughly read and understand the technical details in (Kratsios & Papon 2022) before coming up with more detailed differences between that paper and our work. We also needed some time to think carefully about your comments regarding CNNs.
> > >
> > > -----
> > >
> > > We agree that in both (Kratsios & Papon, 2022) and our work, the input space is transformed to another space for which neural network approximation results already exist, and that the framework in (Kratsios & Papon, 2022) has a similar mathematical expression to ours. We think the main difference between the two papers is the modelling part and the focus.
> > >
> > > In (Kratsios & Papon, 2022), they consider $\phi$ as the feature extraction map, that is a pre-processing step of the data, the features obtained from this step are then used as input of neural networks. The feature map $\phi$ could be designed as fit for each individual application or dataset, but it is not part of the neural network. Then the modelling in (Kratsios & Papon, 2022) is saying that, instead of feeding the extracted features directly into a network, it is more appropriate to apply an inverse exponential map first, to map the data from the feature manifold to a Euclidean space. Then a network can take over from there. Of course, the exponential map would change with input dataset, and it is assumed the exponential map is known or can somehow be calculated from the data.
> > >
> > > In our paper, our focus is compression of the dimension, and our conclusion is that no matter what the data manifold is, it is not necessary to first extract the feature and then estimate the complicated inverse exponential map of that manifold. One can simply apply a single JL map to the data and a network with a low dimensional input can take over. Even if the data does not lie on a manifold (like the set of sparse vectors), the JL map would still work as long as the set has a low complexity. The practical implication is that feature extraction as a pre-processing procedure is not needed, at least for the case of fully connected NNs. For CNNs, the feature extraction via convolution layers probably still useful, but the inverse exponential map can be replaced by the simple JL map. We then went ahead to talk about why common priors such as sparsity and low rankness lead to a low complexity set, making it an end-to-end result for inverse problems. We also think our paper is a bit more friendly to read.
> > >
> > > To summarize, we agree that in essence, the general result in (Kratsios & Papon, 2022) almost reduces to our result when setting $\phi$ to be the JL map and the exponential map to be identity as you said (and thanks for pointing us to that paper), but it is not a natural setting from their modelling perspective. Also, here we argue that this special setting is all that is needed, as it can already compress data from any set or manifold with a low complexity. We think that it is a simple but useful observation to show that the curse of dimensionality for common inverse problems is not that bad.
> > >
> > > -----
> > >
> > > Regarding your U-Net question, indeed, we do not see a place to insert a JL layer into the U-Net architecture, as the JL map would destroy its structure. However, in practice, it common to see a CNN appended by one or more fully connected layers. In this case, the JL layer can be inserted in between the last convolution layer and the first MLP layer. Then our current proof would work, and the same result (reduction of neurons) holds. Also, the same result holds for any MLP layer inserted in other types of networks (ResNet, GNN, etc.).
> > >
> > > -----
> > >
> > > It is true that our CNN result does not make an equivariance assumption on the function. This is because our result is built on (Oono & Suzuki, 2019) which focuses on approximating a more general class of functions which includes both equivariant and non-equivariant functions. As you said, the result in (Zhou 2018) also approximates generic functions. Since this more general class of functions include equivariant functions, then from an approximation point of view, the only benefit of making an equivariance assumption is to possibly derive a tighter universal approximation theorem for this smaller class. However, the only result we're aware of that makes an equivariance assumption is (Petersen & Voigtlaender 2020), but they do not seem to be able to take advantage of this assumption. Their network construction still has a similar number of neurons as needed for approximating the general function class. Hence we feel we should avoid making an equivariance assumption and keep things general if we cannot use it to reduce the number of neurons.
> > >
> > > -----
> > >
> > > Philipp Petersen and Felix Voigtlaender. Equivalence of approximation by convolutional neural networks and fully-connected networks. Proceedings of the American Mathematical Society, 148(4):1567–1581, 2020.

---

### Official Review · Reviewer_MC9E · 2022-10-26

**Confidence:** 3
**Correctness:** 4
**Technical Novelty And Significance:** 3
**Empirical Novelty And Significance:** Not applicable
**Recommendation:** 6

**Clarity, Quality, Novelty And Reproducibility:**

The paper is written clearly and well motivated and the techniques are novel.

Minor comments:
The citation (Iwen et al. accepted (See Arxiv)) could be just changed to it's arxiv link.
Minor typo: Guassian->Gaussian in the last line of page 2.

**Strength And Weaknesses:**

Strengths:
Understanding training w.r.t the low-dimensional embeddings are important and this work has a good characterization of the size of the network required for Lipschitz continuous functions in high dimensions.

Prior works such as (Chen et al. 2019) characterize the number of neurons that depend on the curvature of the low-dimensional manifold in question which can be difficult to estimate. This work sort of alleviates, with a less abstract notion, which are Gaussian widths of the set or the covering number, which seem to be more amenable to estimation from data.

The size of the network grows exponentially in $d$ rather than $D$, the input dimension and the results offer for tradeoffs in practice.

Weakness/Clarifications:
Are there any lower bounds known for the size of the networks? Could the authors comment on the tightness of the size of the networks.

It would have been great to see experiments on some high dimensional datasets as there are usually gaps in approximation characterizations and its training and generalization qualities.

It would be nice to get some bounds on the samples required to for a reasonable approximation, when you estimate the GW or the covering number of the set. Or at least a some discussion pertaining to this.

**Summary Of The Paper:**

Authors in this work study the approximation of high-dimensional Lipschitz functions and characterize the size of the NN required for an arbitrarily good approximation. They are able to say if the data $x \in R^{D}$ is $\rho$-JL embedable in a low-dim manifold $R^d$, then there exists an apporpriate deep network (with say ReLU activation) that can approximate any $L/(1-\rho)$-Lipschitz function $g: [-M,M]^d \rightarrow \mathbb{R}^p$, then there exists a deep network with only a small constant increase in depth, and the size of the network that increases exponentially in $d$ instead of $D$ (the input dimension). They show this by an existence of a JL-embedding for general sets in $\mathbb{R}^D$, in terms of it's Gaussian width (or covering numbers). They use these results to obtain the required size of NN to get an $\varepsilon$-approximation of inverse maps that are Lipschitz continuous.


**Summary Of The Review:**

Overall it is an interesting work on the size of NNs required for approximation of Lipschitz continuous functions in high-dimensions by embedding them in a low-dimensional manifold with complexity parameters that are practical to estimate, as compared to prior work. It would have been great to see some experiments to see this in action.

UPDATE: After the response and the other reviews, it feels like many clarifications were not convincing and thus I decrease my score to 6.

---

> ### Author Response · Authors · 2022-11-16
> **Response to Reviewer MC9E**
>
> There are lower bounds for the required size for a neural network to $\epsilon$-approximate all $C^n$ functions on $[0,1]^d$ with Sobolev norm $\le 1$.
>
> DeVore et. al. (1989) shows that if one imposes the mild requirement that the weight parameters depend continuously on the target function, then at least $O(\epsilon^{-d/n})$ weight parameters are needed.
>
> Yarotsky and Zhevnerchuk (2020) show that with that requirement removed, ReLU neural networks need at least $\widetilde{O}(\epsilon^{-d/2n})$ weight parameters, but there are a few caveats. If the number of weight parameters is restricted to be $\widetilde{O}(\epsilon^{-q})$ for $\tfrac{d}{2n} \le q < \tfrac{d}{n}$, then the required number of layers is at least $\widetilde{O}(\epsilon^{q-d/n})$ and each weight parameter needs to be stored with at least $\widetilde{O}(\epsilon^{q-d/n})$ bits of precision due to the fact that parts of the network employ a bit interleaving/extraction technique to store many Taylor series coefficients in a single weight parameter and then extract the required ones. Only by using at least $\widetilde{O}(\epsilon^{-d/n})$ weight parameters is it possible to use a more practical number of layers and bits per weight parameter possible.
>
> Unfortunately, we are not aware of any lower bounds for the size of universal approximation neural networks for Lipschitz functions on low-complexity sets in high dimensions.
>
> -----
>
> Indeed, we hope to have experiments on some high dimensional datasets in future work.
>
> -----
>
> We plan to add a short discussion on estimating the Gaussian width and/or covering number when revising our work. Thank you for suggesting this.
>
> -----
>
> Ronald A DeVore, Ralph Howard, and Charles Micchelli. Optimal nonlinear approximation. Manuscripta mathematica, 63(4):469–478, 1989.
>
> Dmitry Yarotsky and Anton Zhevnerchuk. The phase diagram of approximation rates for deep neural networks. Advances in neural information processing systems, 33:13005–13015, 2020.

---

> > ### Comment · Reviewer_MC9E · 2022-11-24
> > **Update after author response.**
> >
> > After reading the other reviews and the responses, the authors claimed that prior works such as (Chen et al. 2019) characterize the number of neurons that depend on the curvature of the low-dimensional manifold in question which can be difficult to estimate and this work alleviates, with a less abstract notion, which are Gaussian widths of the set or the covering number. When asked to elaborate, this discussion was not addressed in the response nor added in the pdf.  In addition, the lack of experiments make this less convincing. I would have to decrease my score to 6, in light of this.

---

> > > ### Author Response · Authors · 2022-12-02
> > > **Reply to Reviewer MC9E**
> > >
> > > We apologize that due to unfamiliarity of the program rules, we thought we had until Dec 12th to carefully consider your comments and make revisions. For your comments about Gaussian widths of the set and the covering number, we are happy to provide more elaboration about why then are easier to estimate than the curvature.
> > >
> > > If $\mathcal{S}$ is a continuous domain with a nice mathematical definition (such as the set of sparse vectors, the set of convolutions of two vectors from known low-dimensional subspaces, or the set of low rank matrices), the Gaussian width and/or the covering number can often be directly calculated/bounded as shown in Propositions 8-10. On the other hand, when only samples from $\mathcal{S}$ are available, one may still estimate the covering number. (Kégl 2002) demonstrates a practical method for estimating the intrinsic dimension of a set by using a greedy algorithm to estimate the log-packing number for several different radii $\delta$, and then extrapolating the linear region of the graph of the log-packing number vs. $\log \delta$ to estimate the packing number at finer radii. Then, since the covering number is bounded by the packing number, we obtain bounds for the log-covering number. In contrast, except for 3D manifolds, there are no efficient algorithms to estimate the curvature.
> > >
> > > Again, we apologize for not having been able to respond to this sooner.
> > >
> > > -----
> > >
> > > Balázs Kégl. Intrinsic dimension estimation using packing numbers. Advances in neural information processing systems, 15, 2002.

---

> > > ### Author Response · Authors · 2022-12-05
> > > **Update regarding experiments**
> > >
> > > We recently performed numerical experiments where we trained a neural network to perform sparse recovery. The results show that compressing the measurements with a JL map prior to feeding it into the neural network can actually improve performance. See this link for details. https://anonymous.4open.science/r/ICLR2023_Paper3295-C885/SparseRecoveryExperiment.pdf

---

### Author Response · Authors · 2022-12-02
**Updates to all Reviewers**

We graciously thank the reviewers for their many comments and suggestions. We apologize that due to unfamiliarity of the program rules, we thought we had until Dec 12th to carefully consider the many comments we received and make revisions. While we can no longer upload a revised PDF, we still want to carefully address the questions raised by the reviewers in the responses below to show our appreciation of their hard work. As such, we have posted replies to each reviewer below. Please let us know if you have any further questions or comments.

---

### Decision · Program_Chairs · 2023-01-20

**Decision:**

Reject

**Justification For Why Not Higher Score:**

There was no substantive reason to accept the paper: the mathematical techniques on their own are fairly straightforward and standard; it was also unclear what conceptually the paper teaches us about architecture design or the success of neural networks.

**Justification For Why Not Lower Score:**

N/A

**Metareview: Summary, Strengths And Weaknesses:**

The paper shows representational bounds on expressing functions using a neural network, in the presence of "low dimensional" structure. Specifically, the assumption is the existence of a JL-like embedding: namely, a linear map to a lower-dimensional space, s.t. distances are approximately preserved, and the function in question is a composition with this linear map and a (Lipschitz) function. The reviewers thought that overall the paper is below the acceptance bar: from a purely technical point of view, the proof techniques are rather standard and expected; from a conceptual point of view, it's not clear what the paper teaches us about real trained neural networks (e.g. is it the case that such JL-like embeddings exist, and is this a significant causal factor in the design and success of neural architectures?)